# Wind Load and Wind-Induced Vibration of Photovoltaic Supports: A Review

Bo Nan [1], Yuanpeng Chi [1], Yingchun Jiang [2,*] and Yikui Bai [1]

1. College of Water Conservancy, Shenyang Agricultural University, Shenyang 110866, China; nanbo@syau.edu.cn (B.N.); cyp@stu.syau.edu.cn (Y.C.); baiyikui@syau.edu.cn (Y.B.)
2. College of Engineering, Shenyang Agricultural University, Shenyang 110866, China
* Correspondence: jyclg@syau.edu.cn; Tel.: +86-13889204250

**Abstract:** (1) Background: As environmental issues gain more attention, switching from conventional energy has become a recurring theme. This has led to the widespread development of photovoltaic (PV) power generation systems. PV supports, which support PV power generation systems, are extremely vulnerable to wind loads. For sustainable development, corresponding wind load research should be carried out on PV supports. (2) Methods: First, the effects of several variables, including the body-type coefficient, wind direction angle, and panel inclination angle, on the wind loads of PV supports are discussed. Secondly, the wind-induced vibration of PV supports is studied. Finally, the calculation method of the wind load on PV supports is summarized. (3) Conclusions: According to the particularity of the PV support structure, the impact of different factors on the PV support's wind load should be comprehensively considered, and a more accurate method should be adopted to evaluate and calculate the wind load to lessen the damage that a PV support's wind-induced vibration causes, improve the force safety of PV supports, and thereby enhance the power generation efficiency of PV systems.

**Keywords:** photovoltaic support; wind load; panel inclination angle; wind-induced vibration; photovoltaic system

## 1. Introduction

Environmental problems have become increasingly severe in recent years. Given the goals of "peak carbon dioxide emissions", "carbon neutrality", and compliance with the requirements of sustainable development, PV power generation is a promising renewable energy generation method because of both its environmental protection and economy, and it has been widely developed [1]. China generated 241.4 billion kWh of PV power in 2021 and 325.9 billion kWh in 2022, indicating an annual increase of 35%. However, wind damage to PV supports occurs from time to time, and the most significant load when designing PV supports is the wind load. Therefore, wind resistance is essential for a safe, durable, and sustainable PV power generation system.

There are three modes of support in PV power generation systems: fixed [2], flexible [3], and floating [4,5]. Fixed PV supports are structures with the same rear position and angle. They have the advantages of mature technology, wide application, and simple overhaul and maintenance. In contrast, they face the disadvantage of limited application scope. Meanwhile, a flexible PV panel support is installed on rows of steel cables, which are connected by rigid supports at two ends, realizing a structure spanning 10–30 meters [6]. In addition, external tensile stay cables or internal rigid diagonal supports are used at both ends of the support to reduce the bending moment of the top support at the ends [7,8]. As such, it has the advantages of an extensive and flexible span range, a significant utilization rate of land space, flexible operation, good ventilation performance, and high power generation efficiency [9]. To a certain extent, it can be mutually beneficial regarding crops. A

problem, however, is the immaturity of the technology. A floating PV support is a structure that uses PV panels that are fixed by anchor blocks and floats on the water's surface with a buoy. It not only does not require the construction of a foundation but also adapts to the fluctuations of the water level and helps to achieve the goals of "complementary fishing and light" in water power generation and underwater fish farming [10]. It does not occupy land resources; at the same time, the use of water to cool the PV array can improve the output power and reduce the cost of additional cooling devices for air and water cooling [11,12]. However, floating PV supports at sea level are vulnerable to seawater erosion, and there are few studies on floating PV supports. The specific advantages and disadvantages of the three are shown in Figure 1.

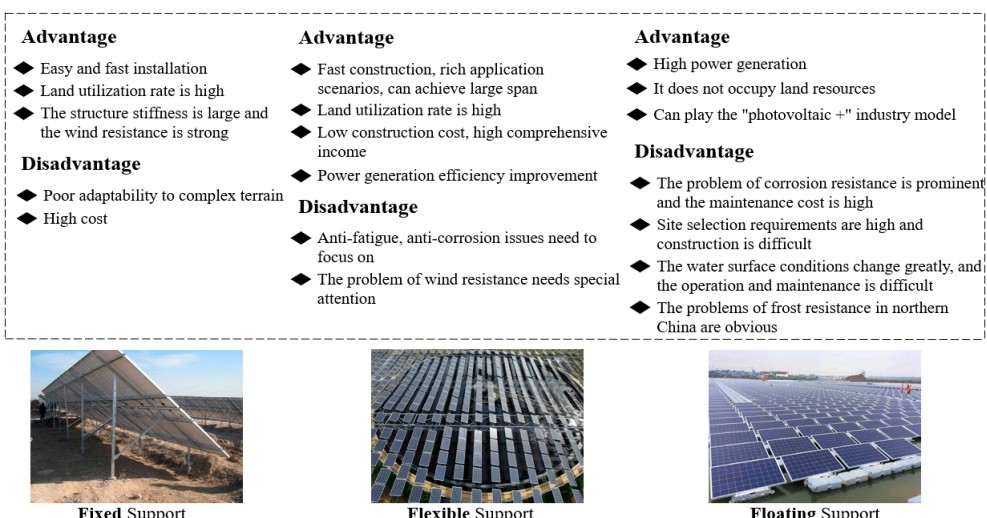

**Figure 1.** Comparison of advantages and disadvantages of three types of PV supports.

Wind often damages PV supports. The wind load is the most significant load considered while designing a PV support [13]. Therefore, wind resistance is essential for a safe and durable PV power generation system. The impact of the wind load on a floating PV support is smaller than that on other PV supports, but regardless of whether fixed or flexible supports are used, the wind load is considered first. Making full use of the previous research results, the following are the main wind load issues associated with the three types of PV supports: (1) the factors affecting the wind loads of PV supports—the main factors are shown in Figure 2; (2) the wind-induced vibration of PV supports; (3) the value and calculation of the wind load of a PV support. It is hoped that this review will provide a reference for the improvement of the wind resistance of PV supports.

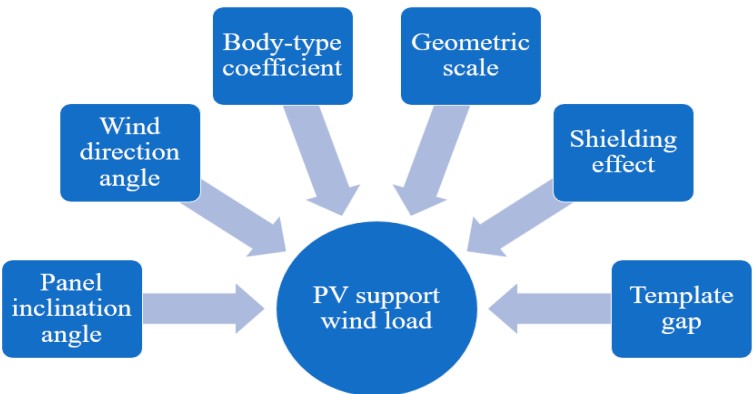

**Figure 2.** Factors affecting wind load of PV support.

## 2. Influencing Factors of Wind Load of PV Panel Support

### 2.1. Panel Inclination Angle

The angle β between the PV panel and the horizontal plane is called the panel inclination (Figure 3). Because of the PV panel's varying inclination angle, a PV power generation system's wind load varies, impacting the system's power generation efficiency.

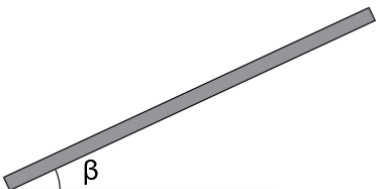

**Figure 3.** Panel inclination angle diagram.

Ma [14,15] et al. investigated the impact of the inclination parameters on the wind load of a PV panel support in a pressure-measuring wind tunnel using rigid PV panel models. The wind load of the PV support was found to be sensitive to the panel inclination angle; in other words, the size coefficient of the PV panel and wind load increased as the inclination angle increased. However, it is not true that the lower the panel inclination angle, the better; when considering the power generation efficiency, the panel inclination angle will have a critical point, but the critical point value remains to be studied. Li [16,17] et al. investigated the influence law of the dip angle and other parameters on the PV wind load on the roof by conducting wind tunnel tests with a rigid scaling model on a single- and double-slope PV vehicle shed (Figures 4 and 5) with a roof dip angle of 20° and 30°. The following conclusions were drawn from their work: (1) the more significant the slope inclination, the greater the wind load value; (2) the windward corner and the leeward ridge sections displayed the highest and lowest wind pressure coefficients of the most disadvantageous measuring location. The greater the slope inclination, the greater the extreme wind suction of the measuring point. The wind distribution of pressure on the surfaces of independent roof panels of flat buildings with modest geometric proportions was investigated by Stathopoulos [18] et al. The test results showed that the panel inclination with the critical wind direction angle of 135°, at which the panels on the back experienced higher suction compared to those in the front, had an impact on the net values of the pressure coefficients corresponding to different configurations.

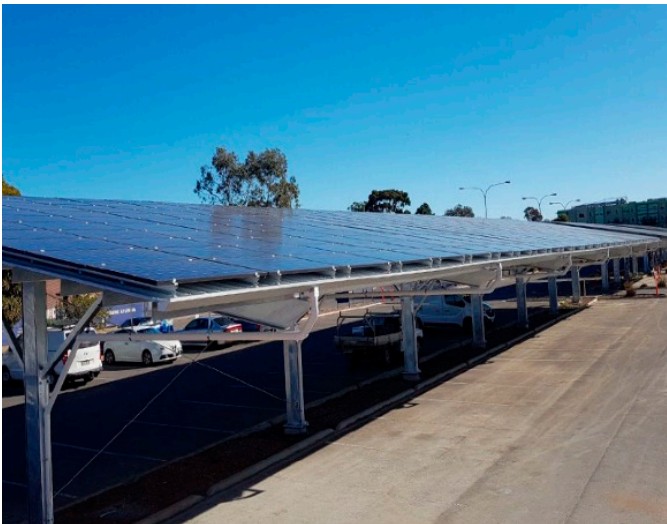

**Figure 4.** Single-slope PV carport.

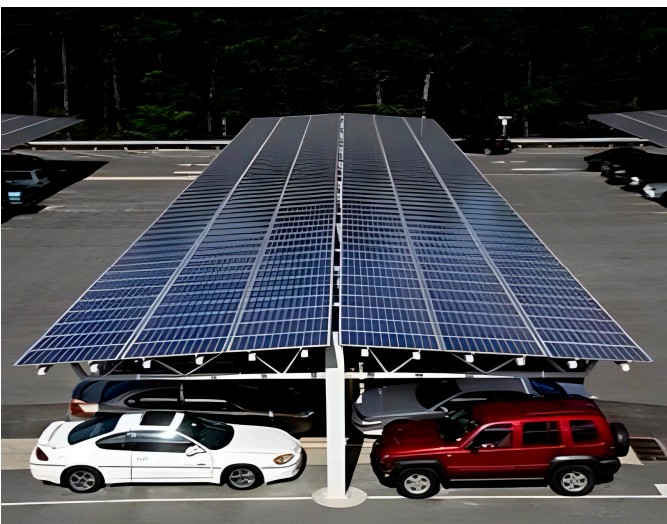

**Figure 5.** Double-slope PV carport.

A large-scale PV system model installed on a residential structure was tested in a wind-wall research facility by Naeiji [19] et al., who concluded that the gap height and building height had little influence on the roof PV support structure, and the wind load value changed by only 5%. The most critical factor influencing the peak pressure coefficients that were produced was the panel inclination angle, and the wind load value changed by 43%. The greater the panel inclination angle, the greater the wind load of the roof PV support structure. Qiao [20] et al. employed computational fluid dynamics (CFD) to examine the PV support array's wind field and suggested regional division methods for varying dip angles and wind load shape coefficients. Kopp [21] et al. performed a wind tunnel study on roof-mounted solar arrays with two panel inclination angles. Two main mechanisms for the aerodynamic loads were obtained: (1) the panels' turbulence; (2) pressure equalization. The array created turbulence at high inclination angles, which raised the net wind loads. Meanwhile, pressure equalization predominated for small inclination degrees.

At Florida International University's Wall of Wind Facility, Naeiji [22] et al. conducted extensive experimental wind testing to examine the effects of geometric parameters on PV systems. The panel inclination angle was the most significant factor influencing wind-induced pressures. The crucial spots with the highest wind pressures were then identified. Jiang [23] et al. examined how the PV array's installation angle and other elements affected the wind load. The body shape and bending moment coefficients of each PV panel rose with the wind direction angle of 30° or 180° when the PV array was installed at a 45° angle. Consequently, the PV array installation angle should not be 45°. Yemenici [24] et al. investigated how the inclination angle affected ground solar panel flow structures, finding that the wind directions and panel inclination angles had a significant impact. The impact of various panel inclination angles on the wind loads of PV supports is summarized in Table 1.

The investigations stated above established that the inclination angle significantly impacted the wind loads of PV supports. It was discovered that the wind load was the most crucial factor when designing PV supports. Future research should concentrate on the sensible arrangement of the PV panel's inclination angles and the improved wind resistance of the PV support system's design. This gives a theoretical foundation for the wind-resistant design of PV panel supports.

**Table 1.** Impact of various panel inclination angles on wind loads of PV supports.

| Author | Angle of Inclination | Conclusion | Reference |
|---|---|---|---|
| Ma et al. | 10°, 30° | The wind load increases as the inclination angle increases. | [14] |
| Li et al. | 20°, 30° | As the roof angle increases, so does the wind load. | [16,17] |
| Stathopoulos et al. | 20°, 30°, 40°, 45° | The panel inclination angle only has an effect in key wind directions. | [18] |
| Naeiji et al. | 20°, 30°, 40° | The factor that most significantly influences the peak pressure coefficient is the angle of panel inclination. | [19] |
| Qiao et al. | 10°, 60° | With varying inclination angles, the wind load's form coefficient changes. | [20] |
| Kopp et al. | 2°, 20° | The array created turbulence at high inclination angles, which raised the net wind load. Meanwhile, pressure equalization predominated for small inclination degrees. | [21] |
| Jiang et al. | 20°, 25°, 30°, 35°, 45° | The installation angle affects the shape coefficient and bending moment coefficient of each PV array. | [23] |
| Yemenici et al. | 25°, 45° | The wind direction and panel inclination have a significant influence on the flow structure. | [24] |

### 2.2. Wind Direction Angle

The wind direction angle, represented by $\alpha$ and having a value range of 0° to 180°, is the angle formed between the wind direction and the PV panel's long-axis direction on the horizontal plane (Figure 6). The wind direction angle significantly influences the wind load on PV supports. For example, distinct wind loads are produced on PV supports at varying wind direction angles. For flexible PV supports, the wind load is highly sensitive when the wind direction angle is 150°. In contrast, the wind load affects fixed PV panel supports and arrays differently.

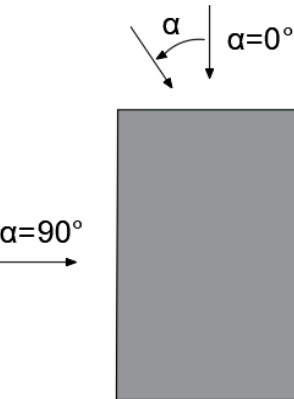

**Figure 6.** Wind direction angle diagram.

Du [25] et al. used the ANSYS 2022R2 finite element software to study the structural wind pressure of a flexible PV support with an increase in the wind azimuth, which refers to the position where the highest absolute value of the wind pressure coefficient gradually moves. Meanwhile, several scholars have analyzed the relationship between the wind angle and fixed PV supports. Using CFD, Huang [26] et al. separately compared the wind load at different azimuthal angles in fixed PV panels and trackers. They discovered that, because of the tracker panels' massive left–right clearance and modest length-to-height

ratios at 45° and 135° wind diversion angles, their wind loads were more significant than those of the fixed panels. Li [27] et al. conducted a numerical wind tunnel simulation for the Hami Chengzi PV system. At wind angles of 90° and 180°, the variation laws of the wind load were similar along the downwind direction; in other words, the windward area had an overall positive pressure, whereas the downwind area had an overall negative pressure. The first two rows of the windward zone had significant variations in the body type coefficient. A rigid model (Figure 7) with a tilt-adjustable PV panel was tested in a wind tunnel by Yin [28] et al. to determine the wind loads on the structure at various wind inclinations and directions. The most sizeable local wind pressure was produced at the windward angle with 45° and 135° wind direction angles, according to the results.

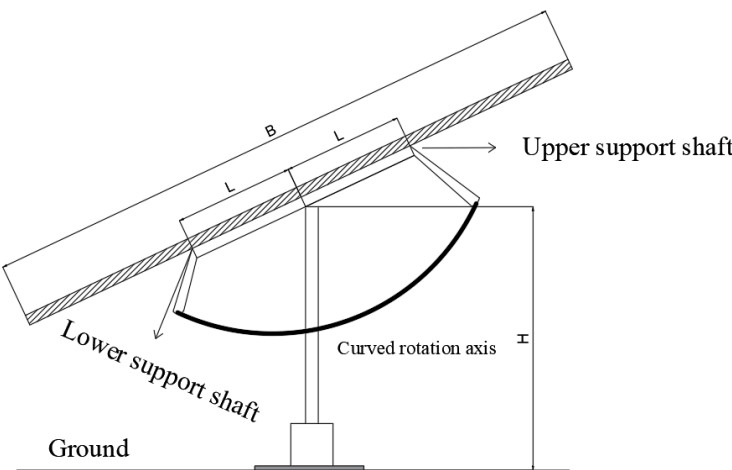

**Figure 7.** PV model with adjustable inclination angle.

Regarding PV arrays, Xu [29] et al. used CFD numerical modeling to assess the wind loads on PV panels at various angles and locations. The wind load variation law was then determined at wind direction angles of 0° and 180°. A thorough study of the wind loads on the array of the surrounding wind field was conducted by Jubayer [30] et al. The third row had the lowest wind loads for wind direction angles of 0° and 180. Shademan [31,32] et al. used three-dimensional Reynolds-averaged Navier–Stokes and CFD simulations to examine the impact of the wind direction angle on PV arrays. The initial findings provided a theoretical foundation for the following wind-resistant design since they demonstrated that the wind load of the structure reached its most significant value at wind direction angles of 0° and 180°. Table 2 summarizes the impact of the wind direction angle on the wind loads of PV panel supports.

**Table 2.** Impact of wind direction angle on wind loads of PV panel supports.

| Author | Wind Direction Angle | Conclusion | Reference |
|---|---|---|---|
| Du et al. | 0°, 180° | The wind pressure on a flexible PV panel support increases as the wind azimuth increases. | [25] |
| Huang et al. | 45°, 135° | The wind load of tracker panels was higher than that of fixed panels owing to the large left–right clearance and the small length-to-height ratio at 45° and 135° wind direction angles. | [26] |
| Yin et al. | 0°, 45°, 135°, 180° | The wind direction angle and dip angle significantly influence the wind load of a PV support. | [28] |
| Xu et al. | 0°, 180° | When the wind angle was 0° and 180°, the second row of the PV panel array had the smallest size coefficient. | [29] |
| Jubayer et al. | 0°, 180° | For 0° and 180° winds, row 3 had the smallest wind load. | [30] |
| Shademan et al. | 0°, 180° | The structure's wind load achieved its maximum value at 0° and 180° wind direction angles. | [31] |

The works mentioned above determined that different wind angles cause the position of the maximum wind load to vary, resulting in different positions for possible damage. Therefore, the design at the position of the maximum wind load should be strengthened to increase the wind resistance, safety, and reliability of PV panel supports.

### 2.3. Body Type Coefficient

The definition of the body type coefficient is as follows:

$$C_{pi} = \frac{P_i - P_s}{P_t - P_s} = \frac{P_i - P_s}{0.5\rho U_r^2} \tag{1}$$

$C_{pi}$ : the average wind pressure coefficient of measuring point $i$ at different wind angles;
$P_t$ : the total pressure at the height of the reference point;
$P_s$ : the static pressure at the reference point height;
$P_i$ : the pressure at point $i$ at different wind direction angles;
$U_r$ : the average wind speed at the height of the reference point.

$$\mu_{si} = C_{pi}\left(\frac{10}{Z_i}\right)^{2\phi} \tag{2}$$

$\mu_{si}$: the body type coefficient at measuring point $i$;
$Z_i$ : the height at point $i$;
$\phi$: the ground roughness index.

The body type coefficient is a significant component influencing the wind load of a PV support. In general, the larger the size factor, the greater the wind load of the PV support. Huang [33] et al. used Fluent to numerically calculate and analyze the surface wind pressure distribution characteristics of PV panel arrays and proposed the body type coefficient for a PV panel group with a wind-resistant design. After researching a single row of PV supports in a wind tunnel, Niu [34] et al. discovered the distribution law of the body type coefficient when the front and rear PV panels interfered. Wang [35] et al. analyzed the wind load distribution of PV panels on a flat roof array by using the wind tunnel test method and concluded that the closer the overall body type coefficient of the PV panels is to the edge of the roof, the greater the wind load that they will bear.

At the same time, some factors will also affect the body type coefficient of the wind load of a PV support and the size of the wind load. Huang [36] et al. studied the wind load distribution on solar PV panels using the wind tunnel test of a rigid model. The experimental results showed that, because of the existence of upstream PV panels, the shape coefficients of the downstream PV panels will decrease to a certain degree. Through a numerical wind tunnel, Huang [37] et al. simulated the body type coefficient of a simplified solar tracker PV panel model. Their research demonstrated that the body type coefficient increased with the gradual increase in the PV panel's elevation angle under each wind direction. In contrast, Wang [38] et al. conducted a wind tunnel test with simultaneous pressure measurement in a uniform turbulent field and used Fluent to numerically simulate and calculate the body type coefficient distribution of the group support under a specific wind direction. The computational findings demonstrated that the test data were compatible with the numerical calculation results and the change rules were consistent when compared to the corresponding results of the wind tunnel test. This gives theoretical justification for the design of wind-resistant PV supports.

The study mentioned above established the significant impact of the body type coefficient on the PV supports' wind load. Therefore, the body type coefficient should be controlled reasonably to increase the wind resistance of the PV panel support system.

### 2.4. Geometric Scale

The geometric scale is a significant issue influencing the wind loads of PV supports. In particular, the wind loads of PV supports with different geometric scales differ. Aly [39–41]

et al. investigated the causes of the variations in peak wind load using a CFD simulation. The findings demonstrated that the geometric size was the primary cause of the various peak wind loads. Using scale models in a boundary-layer wind tunnel, Kopp [42] et al. investigated the wind loads on low-profile roof-mounted solar arrays. The pressure coefficient showed a linear increase with an increasing inclination angle for tilt degrees less than 10°. To investigate the impact of the geometric scale on the wind-induced pressure of rooftop solar panels, Alrawashdeh [43] et al. created, produced, and tested three models in the atmospheric-boundary-layer wind tunnel of Concordia University (Figure 8) with geometric ratios of 1:50, 1:100, and 1:200, respectively. The findings demonstrated the significance of the geometric test scale for solar panel models, particularly when considering the design wind load.

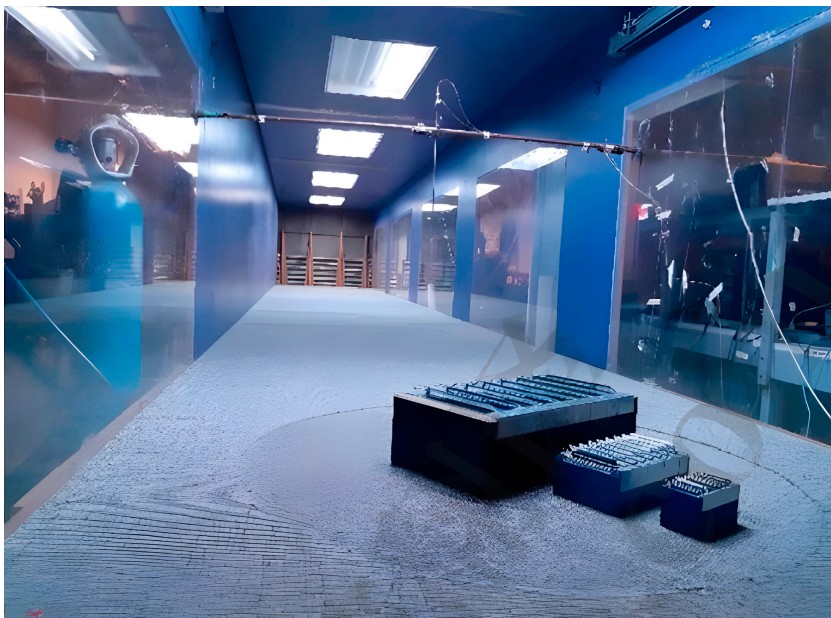

**Figure 8.** Boundary-layer wind tunnel 1:50, 1:100, 1:200 test model.

The works mentioned above establish the importance of the geometric scale in the PV panel supports' wind load. In particular, different geometric scales result in differences in the peak wind load of the PV panel support. In future works, the geometric scale of PV panels should be reasonably designed to strengthen their wind resistance.

### 2.5. Shielding Effect

The shielding effect results in different wind loads at different locations of PV supports. For a numerical simulation and analysis, Fang [44] et al. used ANSYS 19.0 software on PV arrays with a wind angle ranging from 0° to 180°. The simulation result showed that the PV array barrier between the plates impacted the wind load, which led to varying wind loads on the PV panels at various locations. Bitsuamlak [45] et al. examined four test situations to ascertain the impact of wind on independent ground-mounted solar panels. The investigation showed that the wind loads on the neighboring solar panels organized in tandem were significantly decreased by the prominent shielding effect generated by the upwind solar panels. Radu [46] et al. investigated the steady-state wind load characteristics affecting two rectangular flat panel solar collectors of varying sizes through experiments in boundary-layer wind tunnels. Because of the building's and the first row of collectors' sheltering qualities, the wind loads on the solar collectors significantly decreased. The variations in the PV body type coefficients with the inclination angle and panel number were investigated by Lou [47] et al. Upstream PV panels were found to exhibit a notable

shielding effect on downstream PV panels, which remained stable with the number of upstream PV panels.

The shielding effect is inevitable for PV panel arrays. The investigations above demonstrate how the shielding effect affects the wind stress on PV panel supports. To serve as a guide for the design of PV panel supports with wind resistance, future research should reduce the impact of the shielding effect on the wind loads of the supports.

*2.6. Template Gap*

One crucial aspect influencing the wind load of a PV support is the template gap. However, different academics have differing views regarding the influence of the template gap on the wind loads of PV supports; some believe the impact to be quite significant, while others do not.

In a numerical simulation investigation of PV arrays, Ruan [48] et al. concluded that the installation gap of the panels can alter the distribution of wind stress on the upper and lower surfaces of the panels overall. The pressure field on the upper and bottom surfaces of PV panels was investigated by Abiola-Ogedengbe [49] et al. in a wind tunnel. The findings indicated that the template gap would affect the components' surface pressure field. Stenabaugh [50] et al. studied the influence of geometric shapes on the wind loads acting on PV arrays and found that the more significant the gap between panels, the smaller the gap between the panels and roof surfaces and the smaller the net wind loads generated. Yemenici [51] et al. found that the panel gap had a more significant influence on the wind loads of intermediate panels after conducting aerodynamic load measurements on ground-based solar panel arrays. Shademan [52] et al. examined the effects of ground clearance on the average wind load and fluctuating wind loads of solar panels by utilizing the detached eddy simulation method, and the results showed that an increase in clearance would cause an increased average wind load and unstable wind load. Warsido [53] et al. investigated the effects of various spacing parameters on the wind loads of the ground and roof solar arrays in a boundary-layer wind tunnel. They discovered that the wind load coefficient rose as the panel line spacing increased, while the wind load of the roof array decreased as the building edge perimeter spacing increased. Cao [54] et al. carried out several wind tunnel tests to assess the wind stresses on flat roof PV panels. The results showed that the distance between the arrays increased the opposing panel force.

However, according to Wood [55] et al.'s wind tunnel tests, adjusting the panel spacing had a minimal impact on the pressure recorded for solar panel arrays. Geurts [56] et al. conducted wind tunnel experiments (Figure 9) to ascertain the net uplift stress on these systems. In the wind tunnel, they studied the impact of the space between the panel and the roof. According to the results, the effects of the space between the PV system and the roof surface were minor. He [57] et al. conducted numerical simulation calculations on the wind loads of PV panels and drew the following conclusion: in a panel array, the panel gap has little influence on the wind load.

Although a few scholars posit that the template gap has little influence on the wind loads of PV support structures, most scholars still believe that the smaller the template gap and the larger the ground clearance, the greater the wind load generated. Therefore, the influence of the template gap cannot be ignored, and the specific impact of the template gap on the wind load of the PV support structure needs to be further tested and studied. It is also necessary to reasonably increase the template gap and reduce the ground clearance in order to reduce the wind load of the PV support structure, enhance the wind resistance of the PV support structure, and improve the safety and reliability of the PV support structure.

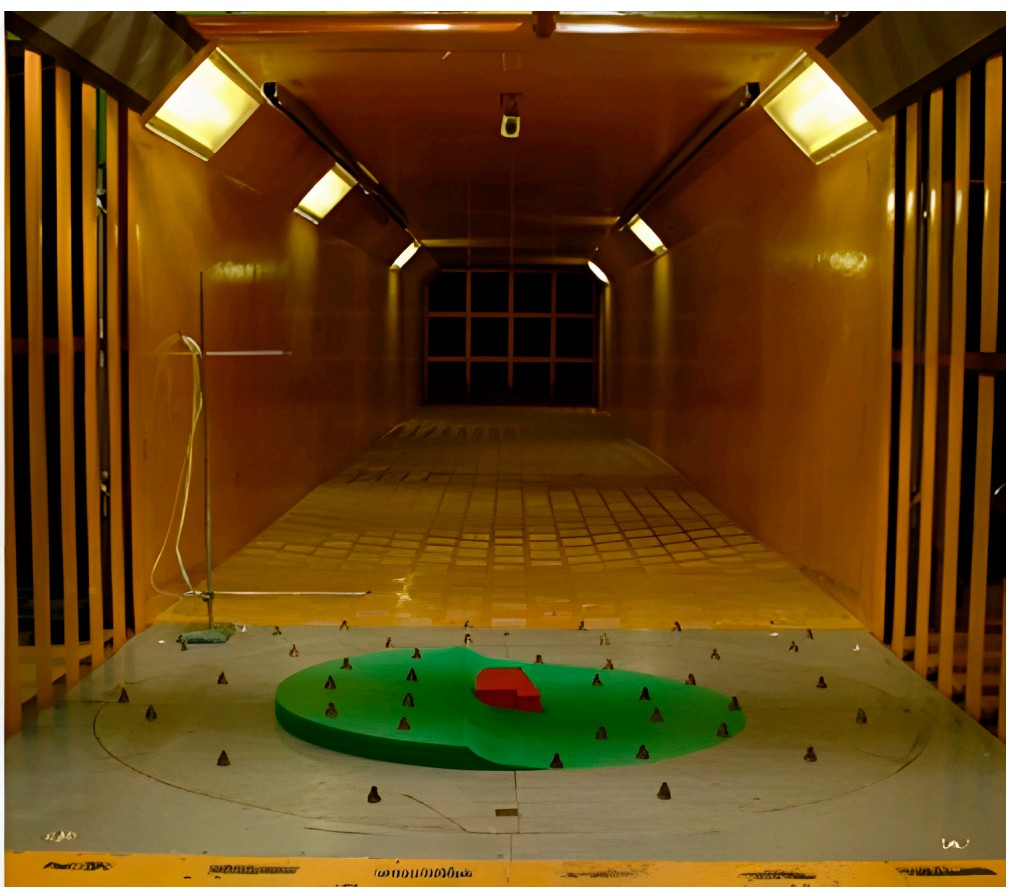

**Figure 9.** BRE wind tunnel view.

*2.7. Other Factors*

It is not only the panel inclination angle, wind direction angle, body type coefficient, geometric scale, shielding effect, and template gap that affect the wind load of the PV support; some other parameters also affect it. Erwin [58] et al. conducted wind tunnel tests on rooftop PV panels to develop a data set of aerodynamic loading effects for rooftop PV systems. The results showed that reasonable agreement was obtained between the WoW and wind tunnel results regarding the lateral and uplift load coefficients for the most critical panel inclination angle of -45° for the flat roof and the 5:12 gable roof. Based on an experimental study, Pfahl [59] et al. concluded that the wind load components varied partly with the panel's aspect ratio. As a result, when organizing the parts of solar trackers, the aspect ratio needs to be considered. Consequently, the effect of the building side was investigated by Wang [60] et al., who conducted wind tunnel studies to study the effects of a constant building height. The results demonstrated a growing trend in the net pressure coefficients for the most significant negative peak area across all wind directions and panels when the inside ratio increased. Ma [61] et al. utilized inflexible models to conduct pressure wind tunnel experiments to investigate the impact of the bottom-flow obstruction ratio on the wind load on the surface of the PV panels. The findings indicated that a bottom-flow blockage significantly enhanced the maximum wind suction on the PV panel, hence decreasing the maximum wind pressure and wind-induced bending moment on the PV panel. Chai [62] et al. conducted several wind tunnel tests (Figure 10) and assessed a stiff panel's pressure to determine the wind pressure coefficients on the PV panel. A wind load model that considered the wind-induced moment was presented based on the nonuniform distribution of wind pressure. This proposed model and its distribution coefficients can be used in designing flexibly supported PV panels.

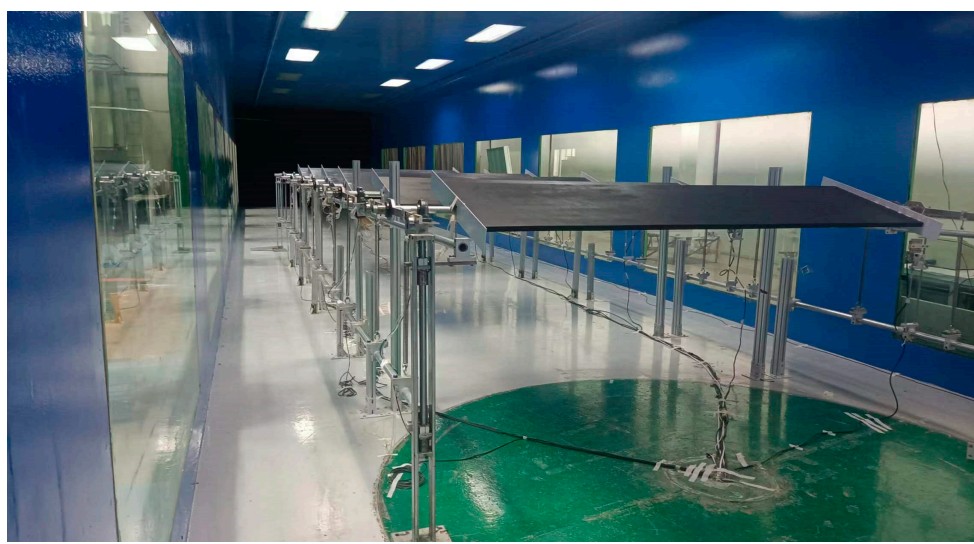

**Figure 10.** Installation drawing of a rigid model wind tunnel.

In order to methodically examine the wind loads on solar panels installed on tall building rooftops, Dai [63] et al. carried out several pressure tests. They concluded that lower-height structures typically produce more pronounced variations in the wind pressure on solar panels. Zhang [64] et al. analyzed the wind load characteristics of near-ground PV arrays shielded by walls, whereby the wall height was found to have a notable impact on the wind load. Browne [65] et al. investigated the impact of parapets on the wind loading of a rooftop solar array with a 10° inclination angle using a scale model wind tunnel test. For many arrays, the parapet effect raised the peak wind load at standard parapet heights. Banks [66] et al. investigated the uplift wind loads on the roofs of wide, rectangular, low-rise, flat-roofed buildings using tilted flat PV panels in an atmospheric boundary-layer wind tunnel. The findings showed a significant difference in wind load between the corner vortices and the cases without them. Pratt [67] et al. conducted a study on the wind patterns surrounding solar arrays that were installed on roofs. Significant vertical gusts caused southern wind peak uplifts at larger inclination degrees, while northern wind peaks were attributed to gusts in the streamwise direction. Wang [68] et al. examined the specific features of wind loads on solar arrays installed on flat roofs with deflectors, as well as those on gable roofs. When comparing gable roofs to flat roofs, the study found that the solar arrays on flat roofs saw more extensive and more rapidly decreasing negative peak net pressures, particularly as the tributary area increased. Aly [69] et al. systematically investigated the wind loads on solar panels installed on various roof zones. This was done in a boundary-layer wind tunnel, considering varied wind directions (Figure 11). The cladding loads on the individual panels either exceeded or fell short of those on the same area of an uncovered roof. To prevent excessive net minimum pressures from affecting the PV panels, it is advisable to steer clear of the sensitive zones of the roof.

When designing PV support systems, the wind load is the primary load to consider for PV power generation. The amount of the PV wind load is influenced by various elements, such as the panel inclination angle, wind direction angle, body type coefficient, geometric scale, shielding effect, and template gap. Therefore, in future works, the reasonable control of the influence of each factor on the PV wind load is necessary to strengthen the wind-resistant design of PV power generation systems.

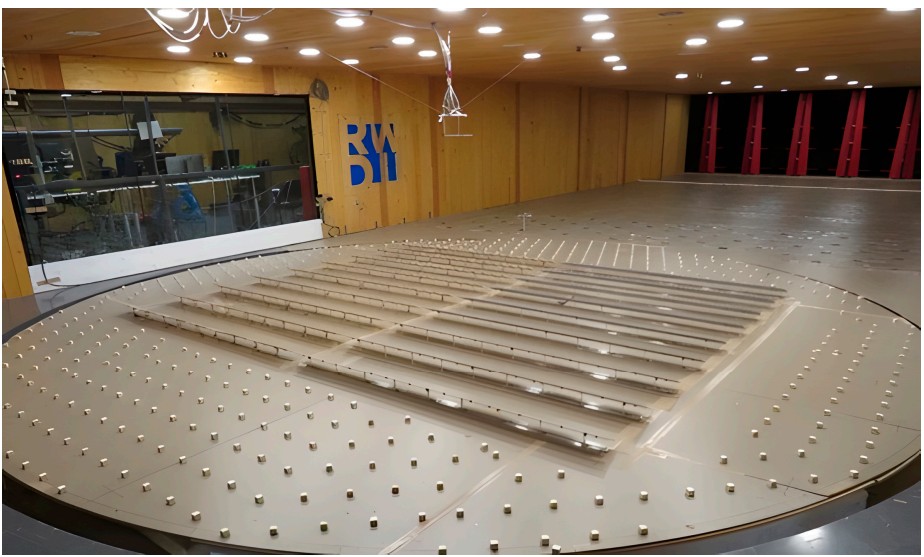

**Figure 11.** Boundary-layer wind tunnel test configuration diagram.

## 3. Wind-Induced Vibration of the PV Support

### 3.1. Wind-Induced Vibration

For PV panels, due to the absorption of solar energy, the temperature may be too high; this is only one of the reasons for the increase in the temperature of PV panels [70], which also reduces the power generation efficiency of PV panels. A wind load accelerates the cooling of PV panels, thereby reducing the cell's temperature and increasing the power generation efficiency for PV power generation. However, the PV panel generates wind-induced vibration due to the wind load, which can damage the system (Figure 12). To solve this problem, a new method has been used to analyze the reliability of solar PV systems.

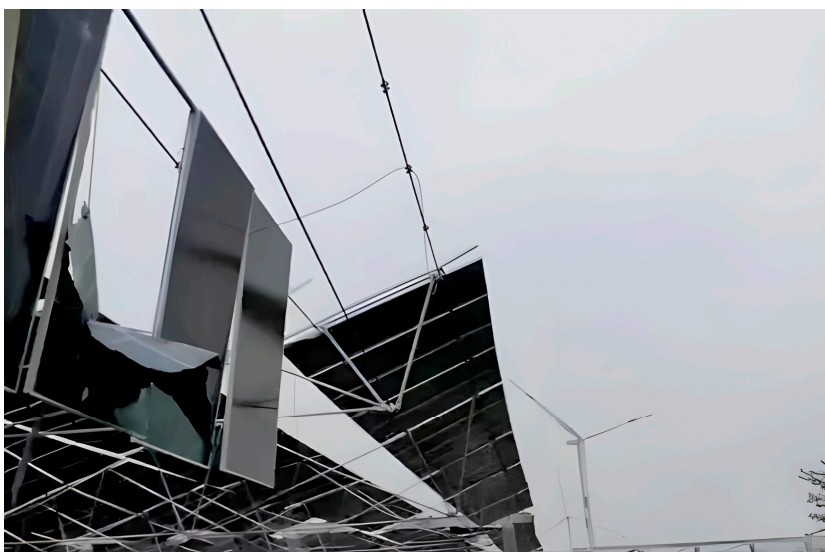

**Figure 12.** Wind vibration damage of PV support.

Ma [71] et al. conducted a wind tunnel test to assess the vibration of elastic models and explored how different parameters affected the wind-induced vibration of tracking solar PV panels. He [72] et al. conducted multiple wind tunnel experiments to examine the vibration properties of PV panels supported by suspension cables. The primary findings can be summarized as follows: cable-supported PV panels are susceptible to significant vibrations when exposed to crosswinds; leeward PV panels experience less vibration

than windward panels, primarily due to the shielding effect. Using CFD simulations, Jubayer [73] et al. looked into the wind loads surrounding a series of solar panels on the ground. A thorough examination of the impact of wind on a solar panel array installed on the ground was achieved by considering both direct and oblique wind directions. In order to investigate the changes in the wind-induced vibration of PV panels, considering the wind speed, Li [74] et al. tested elastic-suspension segmental models with varying PV panel inclinations in wind tunnels. The flexible PV segmental models' flutter critical wind speed values at different inclinations were also found. Wang [75] et al. used an arrangement involving two diagonal tie bars and one vertical rigid bar to analyze the modal and maximum displacement curve under wind–pressure time history excitation and investigated the control effect of different vibration control methods on multirow flexible supports. The results showed that the vertical rigid rod arrangement could effectively coordinate the displacement of the upper chord, middle–upper chord, middle–lower chord, and lower chord at different survey points across the middle and simultaneously reduce the inconsistencies concerning the displacement of the strings of multiple rows of flexible PV supports. Zhao [76] et al. analyzed the displacement state of a structure using two analysis methods with and without consideration of the spatial correlation of the pulsating wind, in order to investigate the suspension cables of a long-span support system accurately. The displacement states of the upper and lower chords of the structure were investigated. For structures exhibiting a large span and long space, the spatial correlation of the pulsating wind may not be considered, adversely affecting the analysis results. Consequently, an envelope design for a large-span stent system was recommended. A response history analysis was employed by Schellenberg [77] et al. to examine how solar panels affect wind (Figure 13). According to the response history study, code-design-level winds under uplift can be withstood by a flexible solar array support system with a sufficient ballast weight or attachments, especially at the edges and corners of the array, and suitable structural connections.

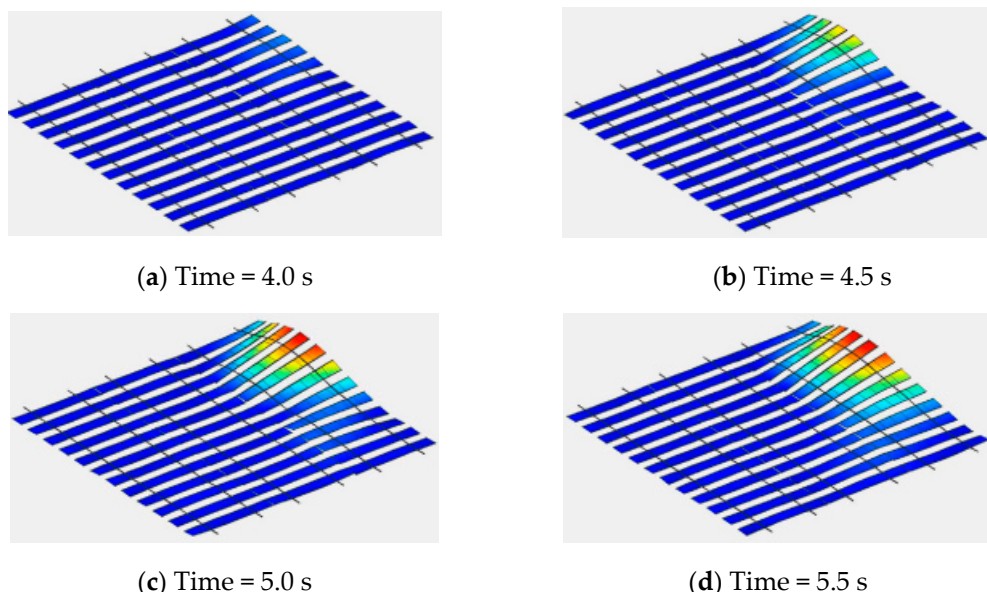

(**a**) Time = 4.0 s

(**b**) Time = 4.5 s

(**c**) Time = 5.0 s

(**d**) Time = 5.5 s

**Figure 13.** *Cont.*

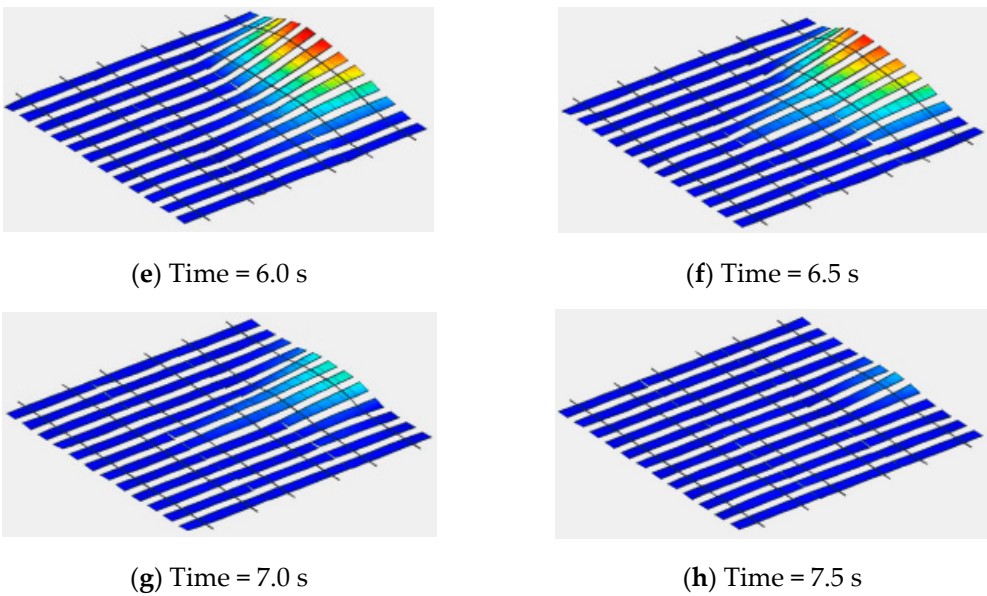

(**e**) Time = 6.0 s  (**f**) Time = 6.5 s

(**g**) Time = 7.0 s  (**h**) Time = 7.5 s

**Figure 13.** Response-history analysis chart.

Yuan [78] et al. used a dynamic analysis method to simulate the dynamic response of a PV steel panel support under strong winds. A new calculation method for the design of PV steel structures and a basis for the study of their dynamic performance and structural optimization was provided. Xu [79,80] et al. used the finite element program ABAQUS to investigate the wind-induced vibration coefficient of a fish-belly PV cable truss structural support. The numerical analysis results of the dynamic wind load obtained a large discrete-type displacement–wind–vibration coefficient, suggesting the practicality of using stress–wind–vibration coefficients. In an experiment, Gong [81] et al. examined a rigid heliostat model subjected to three-dimensional wind loads in a wind tunnel. The findings of the measurements made by Peterka et al. were compared with the maximum wind force coefficients that were obtained and the corresponding wind directions. Based on the results of the analysis, a desirable stow location was recommended to withstand wind loads. Tamura [82] et al. described the wind-induced vibration of a solar wing system obtained in wind tunnel experiments and investigated its aeroelastic instability using a scaled model. In order to investigate the flow characteristics surrounding solar arrays installed on a flat roof building (Figure 14) for two typical wind directions and elucidate the relationships between the flow field and wind pressure distributions on solar panels, Wang [83,84] et al. conducted extensive eddy simulations. It was concluded that the PV panel was more prone to vibration when its back was facing the wind. Liu [85] et al. studied a typical solar panel structure while considering an equivalent static wind load. A reference for the design of solar energy structures was thus provided by comparing the displacement and stress of the structure under various working conditions and determining the impact of the wind load on the structure.

Future research should lessen the effect of the wind load on the wind-induced vibration of PV power generation systems, consequently increasing the efficiency of PV power generation systems, to address the detrimental effect of wind loads on panel PV supports.

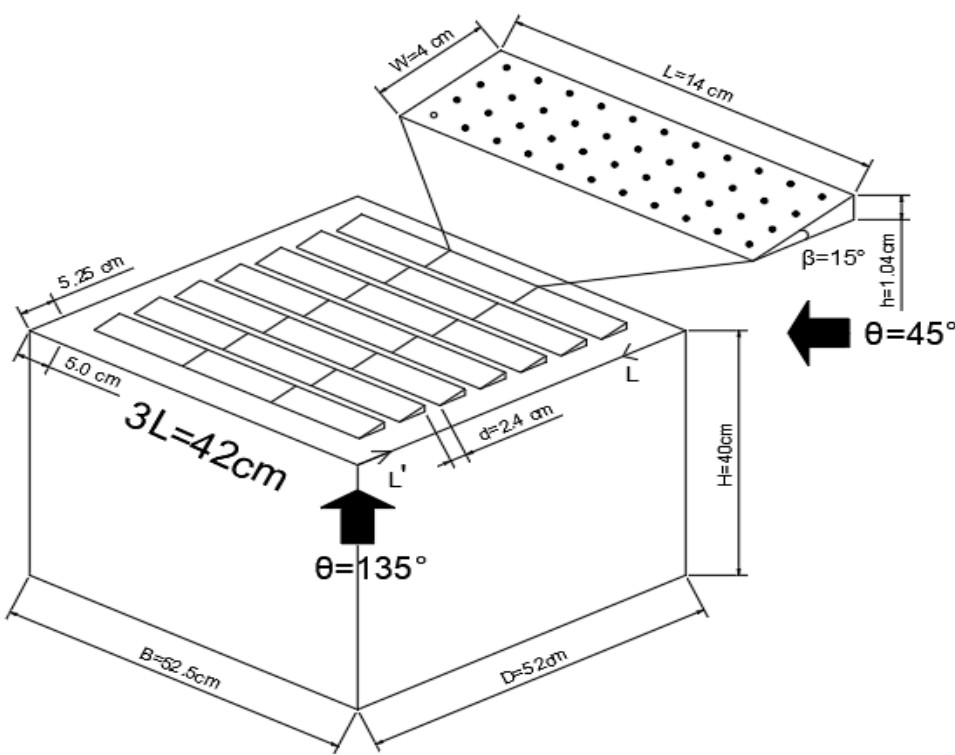

**Figure 14.** Schematic view of flat-roof-mounted solar arrays.

### 3.2. Example Analysis

According to the comparison of the advantages and disadvantages of fixed PV supports, flexible PV supports, and floating PV supports in Figure 1, the scope of application of fixed PV supports and floating PV supports is relatively limited; a fixed PV support is only suitable for flat terrain, and a floating PV support is only suitable for flat or sea level water, while the flexible PV support has a broader range of application. It is applicable in harsh terrain but is affected by the wind load; a flexible PV support makes it easy to produce wind-induced vibration. Thus, an analysis is conducted on a flexible PV support's wind-induced vibration.

#### 3.2.1. Numerical Model

Firstly, a refined model of the flexible PV support was established (Figure 15). The model comprises a horizontal load-bearing structure, beam structure, cable structure, tripod structure, and PV panel. Because of the three horizontal load-bearing structures, three models, such as a horizontal load-bearing structure of the cable, an inclined steel column horizontal load-bearing structure, and a horizontal force-bearing structure with eight inclined steel columns, are generated. In this paper, the span of the flexible PV support is 8 m, and the height is 2.6 m. In the horizontal load-bearing structure, the base size is 500 mm × 500 mm × 500 mm and the column structure's size is 200 mm × 200 mm × 2400 mm; the beam structure's size is 2000 mm × 200 mm × 200 mm; the cable structure's length is 8 m; the tripod's size is 1420 mm × 820 mm × 200 mm; and the PV panel structure's size is 1640 mm × 992 mm × 48 mm.

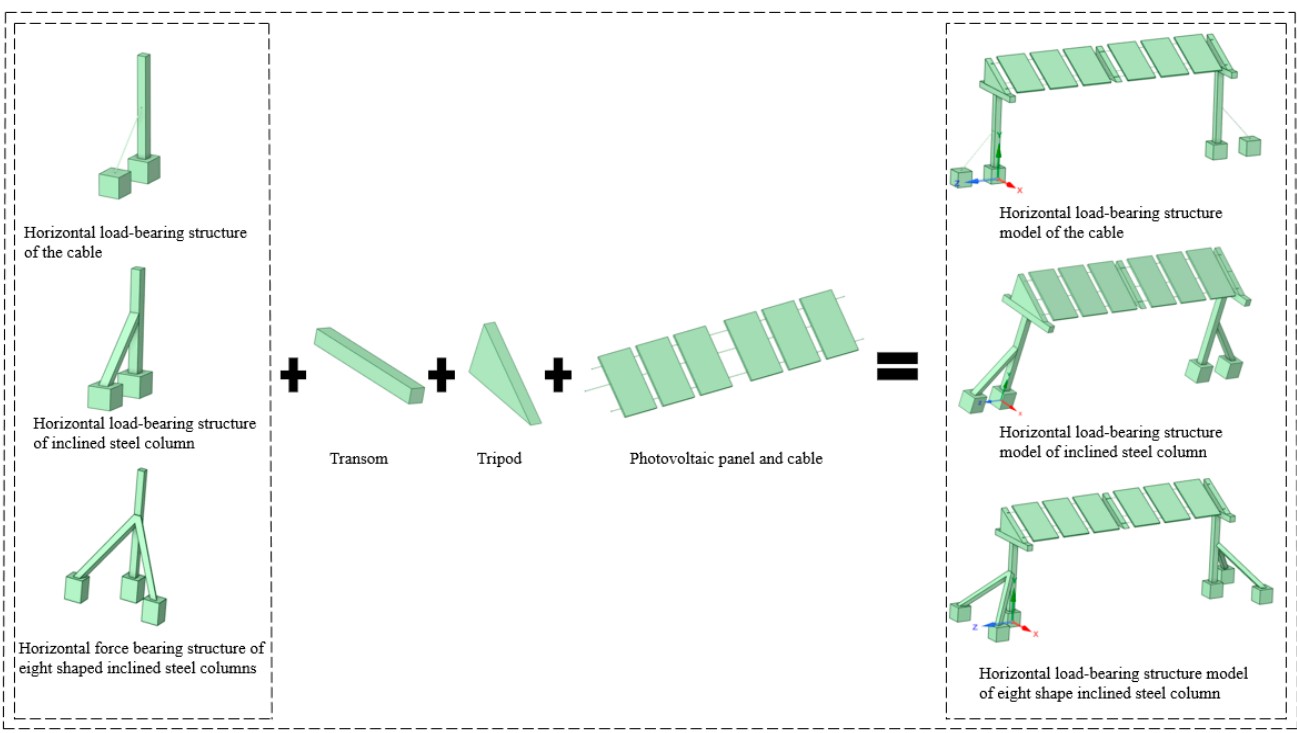

**Figure 15.** Fine model of flexible PV support.

### 3.2.2. Material Parameters

In this simulation experiment, concrete material is used as the foundation. The columns and beams in the horizontal load-bearing structure are all made of structural steel. The tripod structure adopts aluminum alloy material; the PV panels are made of PV materials; each material is an isotropic elastic material; and all components are assigned corresponding material properties (each material's properties are shown in Table 3).

**Table 3.** Material characteristics.

| Material Name | Density (kg/m$^3$) | Young's Modulus (Pa) | Poisson's Ratio |
|---|---|---|---|
| Concrete | 2300 | $3 \times 10^{10}$ | 0.18 |
| Structural steel | 7850 | $2 \times 10^{11}$ | 0.3 |
| Aluminum alloy | 2770 | $7.1 \times 10^{10}$ | 0.33 |
| PV material | 2600 | $1.1 \times 10^{11}$ | 0.225 |

### 3.2.3. Simulation Results

This paper uses the ANSYS finite element analysis software to load wind loads on the three horizontal load-bearing structural models. The wind load direction is horizontal, and the wind load value is constantly increased to extract the ultimate wind loads of the three horizontal load-bearing structural models. The final simulation results are shown in Figure 16.

The maximum displacement of the three horizontal load-bearing structural models when they reach the ultimate bearing capacity is around 12.4 mm, as seen by comparing the above three simulation results. Therefore, to ensure safety, the displacement should be limited to 12.4 mm. At this time, the ultimate wind loads of the three models were extracted and compared (Figure 17).

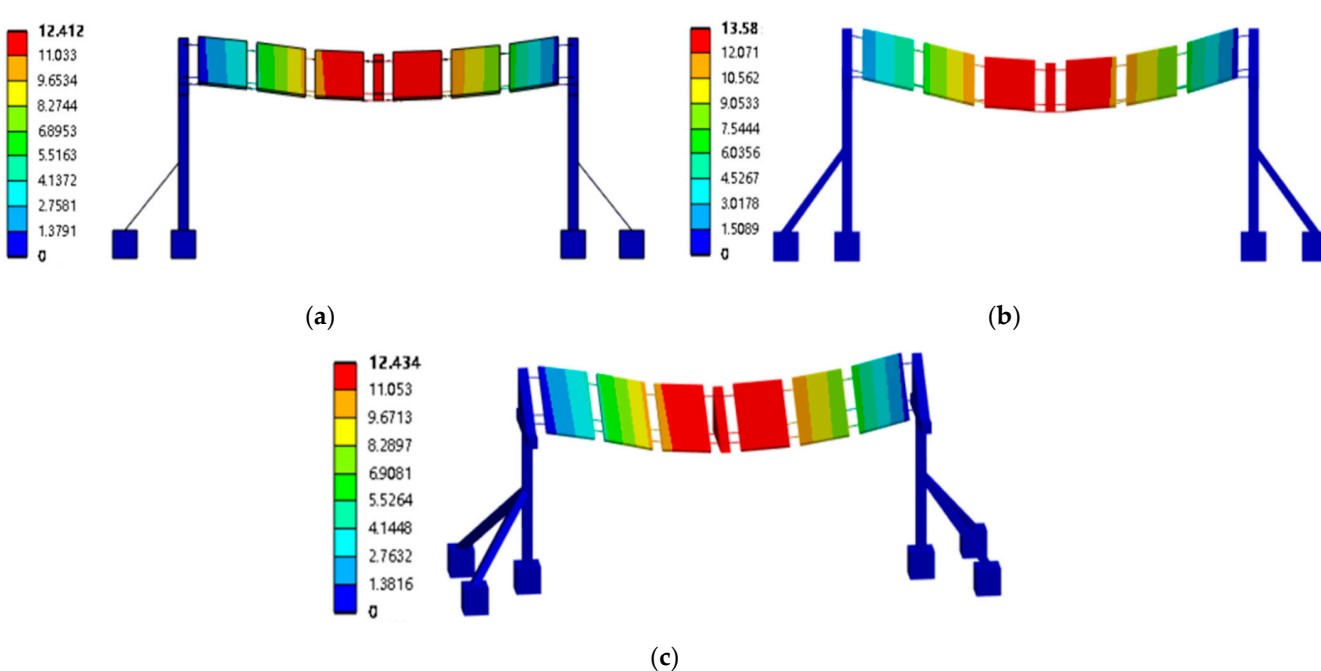

**Figure 16.** Comparison of the total deformation of the three models. (**a**) Maximum displacement of the cable model. (**b**) Maximum displacement of the inclined steel column model. (**c**) Maximum displacement of the model with eight inclined steel columns.

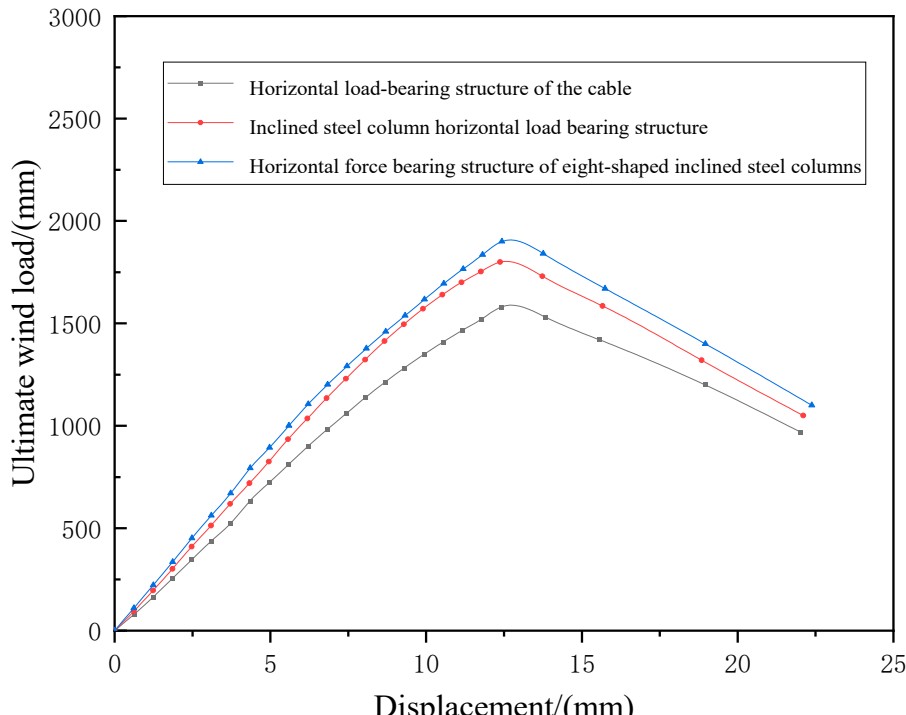

**Figure 17.** Comparison of the ultimate wind loads of three models.

Three types of ultimate wind loads are obtained through the concrete analysis of the three types of flexible PV support structure. Through the comparative analysis of these three wind loads, it is concluded that the ultimate wind load of the horizontal load-bearing structure of the cable is the smallest, the ultimate wind load of the horizontal force-bearing structure with eight inclined steel columns is the largest, and the ultimate wind load of the inclined steel column horizontal load-bearing structure is similar to that of the horizontal

force-bearing structure with eight inclined steel columns. Therefore, the horizontal force-bearing structure with eight inclined steel columns and the inclined steel column horizontal load-bearing structure are more reliable. However, compared with the inclined steel column horizontal load-bearing structure, the horizontal force-bearing structure with eight inclined steel columns is more complicated, and the cost is higher, so the inclined steel column horizontal load-bearing structure is safer, more reliable, and economical, and the loading characteristics of the inclined steel column horizontal load bearing structure need to be further studied. In summary, the inclined steel column horizontal load-bearing structure is the best choice to reduce the wind vibration of the flexible PV support.

3.2.4. Discussion

The wind load is a vital load affecting PV supports, and the harm caused by wind-induced vibration due to wind loads is enormous. Aiming at the wind-induced vibration of flexible PV supports, a PV building integration technology [86,87] was proposed to reduce the harm caused by wind vibration. PV building integration (Figure 18) is a technology that integrates solar power generation products into buildings. Because of this characteristic, it offers a measure to avoid wind-induced vibration during PV power generation. However, at the same time, its application range is also relatively narrow, only some buildings are suitable for PV building integration, and it will also produce a series of problems, such as temperature problems [88].

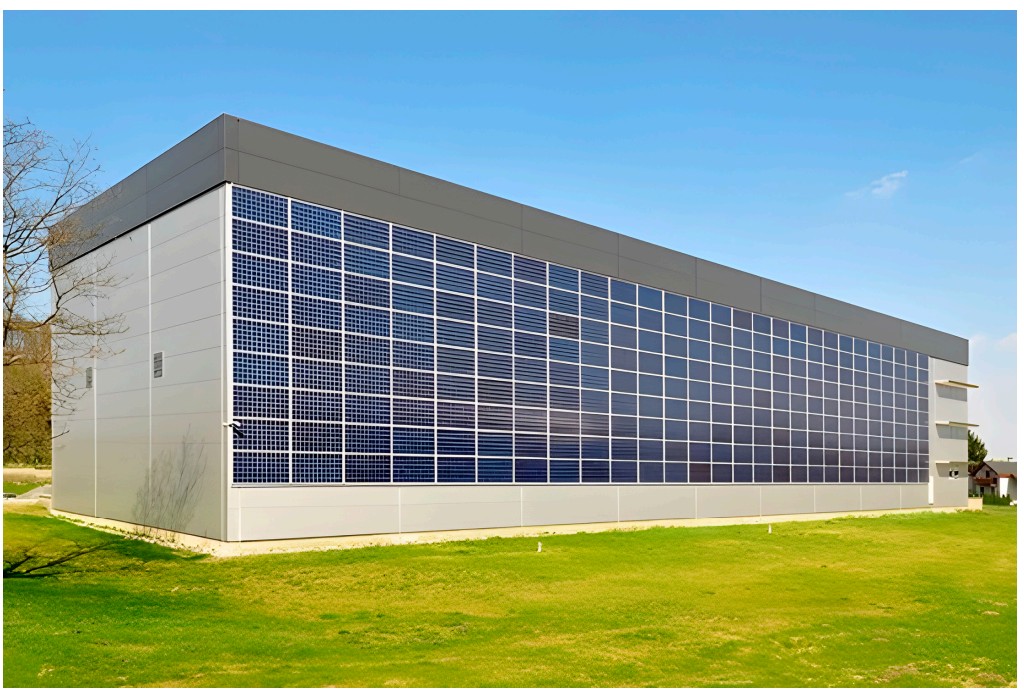

**Figure 18.** PV building integration project.

There is also a unified active roof wind design method [89] that defines the fixtures of these products and suggests rules for the design of substructures against wind loads, but, so far, this technology has yet to be developed.

## 4. Calculation of Wind Load of PV Panel Support

The wind load is the most significant load when designing a PV support; thus, its value and calculation should be investigated. Different countries have their own specifications and, consequently, equations for the wind loads of PV supports. Zou [90] et al. compared the main parameters of the wind load in relevant local and international regulations used for the wind load calculation of the fixed supports of PV structures. The findings demonstrated that different regulations' restrictions on the wind load on PV installations

fluctuate significantly, leading to various wind load values. Two recognized techniques for the determination of wind loads on structures—such as solar panels—were introduced by Banks [91] in the United States. Meanwhile, Zhao [92] et al. conducted a comparative analysis of the most widely used Chinese, American, European, and Japanese codes for PV wind load calculation methods. In particular, the similarities and differences in the wind load calculation factors and their correction coefficients for each code were compared and investigated. The Chinese, American, European, and Japanese codes were noted to consider the ground roughness category, topographic conditions, wind pressure height variations, and wind vibration coefficients, but there were differences in the correction coefficients used for different countries. Thus, He [93] et al. proposed a new wind load distribution model for PV arrays (Figure 19) based on the comparison between the Japanese and Chinese codes and combined this with the results of wind tunnel experiments for a more accurate evaluation of the wind load on a PV array and for the determination of the actuator axial force of tracking systems.

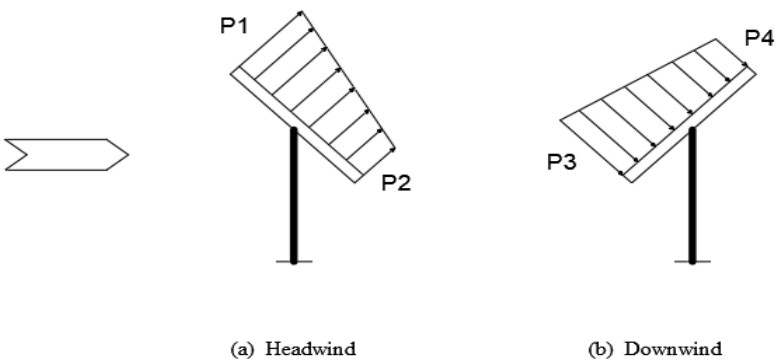

(a) Headwind          (b) Downwind

**Figure 19.** A new wind load distribution model for PV arrays.

Wind tunnel studies were conducted by Kopp [94] et al. to determine the design wind load for rooftop solar arrays. Meanwhile, based on a compilation of wind tunnel experiments carried out by RWDI using rigid pressure models, Browne [95] et al. presented a succinct method to determine the design wind loads for multi-row ground-mounted solar arrays, including both static and dynamic wind load coefficients compatible with ASCE 7. Yang [96] et al. established a wind load calculation model for tracking supports based on the national standard analysis of the heterogeneous loads of PV panel supports. This method offers clear direction for the PV tracking support's optimization. Further, the developed MATLAB 2021b calculation tool can conveniently and efficiently calculate the PV wind load. Three wind load models, namely the uniform distribution, trapezoidal distribution, and eccentric moment models, were developed by Ma [97] et al. in terms of the structural features of a solar panel. Gao [98] et al. used computational calculations and wind tunnel testing to investigate the wind field properties of a PV panel support unit. The outcomes demonstrated that the PV panel's wind load influence variables were parameterized. Additionally, formulas for wind loads were derived together with examples, providing a guide for the design of wind-resistant PV structures. Zhang [99] et al. used Labview and a pull-pressure sensor to analyze the pulling and pressing on PV panels. Subsequently, the results were compared with those of previous numerical analysis and verification experiments, providing the basis for the strength design of PV panels and their tipping moment calculation. The wind load of floating PV arrays (Figure 20) was simulated by Chen [100] et al. by CFD. The variation rule of the wind load under different directions was obtained to verify the accuracy of the design specifications. According to the results, the first row of upwind PV arrays was primarily affected by the wind load. Consequently, improvement measures were established based on the results. Thus, this study is essential for the design and optimization of PV panel arrays and the correct selection of the wind load in the anti-wind design of solar panels. Han [101] et al. utilized a rigid model wind tunnel test of the wind load distribution of a single set of ground-mounted PV panels.

Subsequently, recommended values were obtained for the test operation of a single set of PV panels, a PV panel array, and a rooftop PV panel with different parapet heights. The detailed calculation formula for the wind load is shown in Table 4.

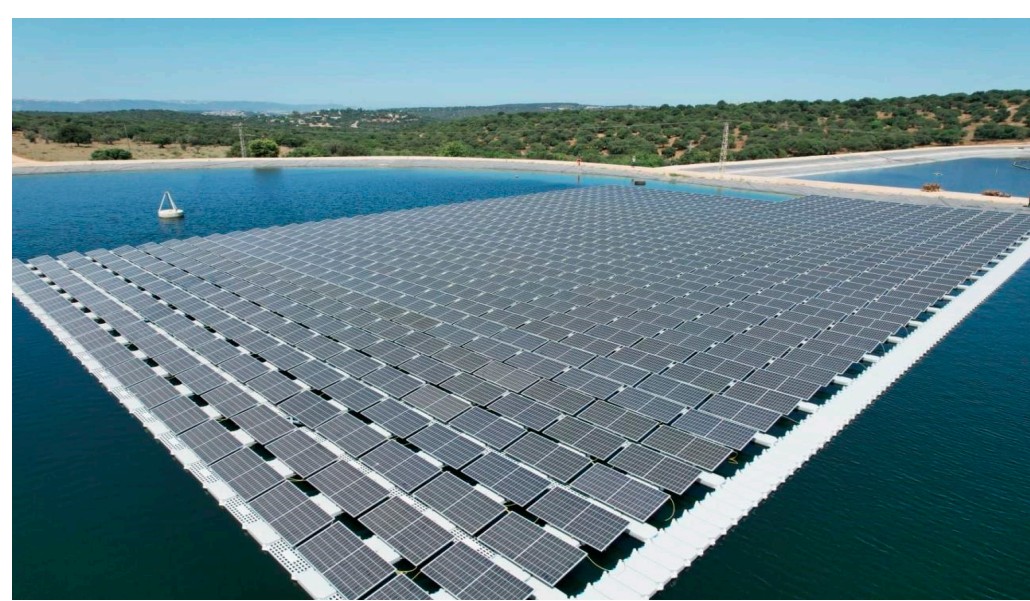

**Figure 20.** Floating PV array.

**Table 4.** The calculation formula for the wind load.

| Author | Formula | Characteristic | Reference |
|---|---|---|---|
| Zou et al. | $W = w_k A_w, w_k = \beta_z \mu_s \mu_z w_0$ | The dynamic effect of the wind must be considered. | [90] |
| Zhao et al. | $p = q_h G C_N$ | Short time interval and long recurrence period. | [92] |
| He et al. | $W = \frac{(w_{k1}+w_{k2})}{2} A_w$ | Close to the actual situation, conducive to the realization of PV structures with complex designs. | [93] |
| Browne et al. | $P_{\text{total}} = \overline{P}_{net} \pm \widetilde{P}_{net} \sqrt{\left(g_{B\_P}\right)^2 + (g_{R\_P})^2 \left(\text{DAF}_P^2 - 1\right)}$ | It is assumed that the wind deflection is slight and therefore the structure is inflexible. | [95] |
| Yang et al. | $W_k = \frac{1}{3} w_{k1} + \frac{2}{3} w_{k2}$ | Computational models are more realistic. | [96] |
| Gao et al. | $P = c_{pmean} \mu_z \varphi_p \varphi_\beta \varphi_\alpha w_0 A$ | It provides a reference for the wind-resistant design of solar PV structures. | [98] |
| Zhang et al. | $W = 0.856 V^2$ | The fitting values are consistent with the empirical values. | [99] |

## 5. Conclusions

(1) For PV support structures, the most critical load is the wind load; the existing research only focuses on the panel inclination angle, wind direction angle, body type coefficient, geometric scale, shielding effect, template gap, and other single factors that impact the wind loads of PV support structures. Future work should consider the comprehensive impact of the above factors for the wind-resistant design of PV support structures.

(2) The wind-induced vibration caused by wind loads is one of the main reasons for the failure of PV supports, so the research focus is not only to improve the power

generation efficiency of PV systems but also to reduce the wind-induced vibration of PV support structures.

(3) The wind load is an essential load in the PV support structure, so determining the wind load is critical. The existing wind load calculation formulas for PV support structures have their limitations. In the future, the wind load calculation formulas of PV support structures should be further improved based on their predecessors to better achieve the wind-resistant design of PV support structures.

**Author Contributions:** Conceptualization, Y.J. and B.N.; validation, Y.J. and Y.C.; writing—original draft preparation, Y.J., B.N. and Y.C.; writing—review and editing, Y.J., B.N. and Y.C.; visualization, Y.B. and Y.J.; supervision, Y.J.; project administration, Y.J.; funding acquisition, B.N. and Y.J. All authors have read and agreed to the published version of the manuscript.

**Funding:** Basic Scientific Research Project of Colleges and Universities of Liaoning Province (LJKZ0698); Scientific Research Funding Project of Liaoning Province, China (Grant number: LJKZ0689).

**Institutional Review Board Statement:** Not applicable.

**Informed Consent Statement:** Not applicable.

**Data Availability Statement:** The data presented in this study are available on request from the corresponding author.

**Conflicts of Interest:** The authors declare no conflicts of interest.

## Abbreviations

| Notation | Description |
| --- | --- |
| $W$ | the design wind load |
| $\omega_k$ | the standard value of the wind load |
| $\omega_0$ | the basic wind pressure |
| $\mu_s$ | the body type coefficient of the wind load |
| $\mu_z$ | the wind pressure height coefficient |
| $\beta_z$ | the wind vibration coefficient |
| $A_W$ | the wind area |
| $p$ | the design wind pressure |
| $G$ | the gust coefficient |
| $C_N$ | the body type coefficient of the wind load |
| $q_h$ | the velocity of the wind pressure at a given altitude |
| $P_{total}$ | the total wind pressure |
| $\overline{P}_{net}$ | the mean wind pressure |
| $\tilde{P}_{net}$ | the standard deviation of the wind pressure |
| $g_{B\_p}$ | the resonant peak factor |
| $g_{R\_p}$ | the resonant peak factor |
| $DAF_p$ | the dynamic amplification factors associated with the normal force |
| $W_k$ | the design wind load |
| $c_{pmean}$ | the average wind pressure coefficient |
| $\mu_z$ | the height coefficient |
| $\varphi_p$ | the position coefficient |
| $\varphi_\beta$ | the wind direction angle factor |
| $\varphi_\alpha$ | the dip coefficient |
| $A$ | the wind area |
| $V$ | the wind speed |

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
