# Peer review of "Wind Load and Wind-Induced Vibration of Photovoltaic Supports: A Review"

_sustainability, doi:10.3390/su16062551_

Round 1

Reviewer 1 Report

Comments and Suggestions for Authors

The overall language of the article is smooth, and a good summary of the achievements of flexible photovoltaic supports in recent years. It is suggested to be published after minor modifications.

1. The abbreviations in the text should be consistent, such as "photovoltaic" in lines 445 and 449 without an abbreviation; the full text has abbreviated "photovoltaic" to "PV".

2. It is encouraging for review articles to make some analysis based on their own research results, but the conclusion of 3.2.3 example analysis is relatively weak, which is suggested to be reflected in the conclusion and carried out in subsequent studies.

3. The conclusion part should be more refined.

Author Response

Thank you for your approval. We have carefully revised this manuscript according to your suggestions.

Comment 1. The abbreviations in the text should be consistent, such as "photovoltaic" in lines 445 and 449 without an abbreviation; the full text has abbreviated "photovoltaic" to "PV".

Response: Thanks very much for the kind suggestions. We have abbreviated "photovoltaic" to "PV" in its entirety and remain consistent throughout the text.

Comment 2. It is encouraging for review articles to make some analysis based on their own research results, but the conclusion of 3.2.3 example analysis is relatively weak, which is suggested to be reflected in the conclusion and carried out in subsequent studies.

Response: Thanks very much for the kind suggestions. We have perfected the conclusions in 3.2.3 examples and marked them in red in this paper, and finally put forward suggestions for subsequent research.

Comment 3. The conclusion part should be more refined.

Response: Thanks very much for the kind suggestions. We have streamlined and improved the conclusions so that each conclusion is focused on the main points.

Reviewer 2 Report

Comments and Suggestions for Authors

In order to understand the state-of-the-art in this field in a better way, the most relevant background information should be provided, and the most relevant literature should be cited. Especially, to give the readers a sense of continuity, a literature check of the papers published in this in recent years is preferred, and the content of relevant papers should be related to the results and findings presented in your publication.

1.       Please clarify the equations in detail.

2.       Please check the format of the reference.

3.    The description of the equations in this manuscript is not clear. That is, the definition of symbols and abbreviations in equations is not clear enough. Please revise and define it.

4. Some sentences with long descriptions are difficult to read and understand. Some typographical and grammatical errors of this manuscript should be carefully checked and corrected.

5.   Can the author clarify to express what contribution to the industry sector and academic field with this research?

Comments on the Quality of English Language

Please check the Comments file.

Reviewer 3 Report

Comments and Suggestions for Authors

The article submitted for review is of a review nature. It posits the thesis that one of the main causes of failure in photovoltaic module supports is the wind load. An attempt is made in the article to analyze this problem for three types of photovoltaic module installations: fixed support, flexible support, and floating support. The authors, in an attempt to prove the thesis, cite conclusions from the results of studies in already published articles. However, these conclusions are very general and do not bring the reader closer to expanding their knowledge.

For example:

  1. Lines 78-81, the authors cite the article by Ma et al., from which it follows that the risk of damaging PV module supports due to wind pressure increases with the angle of inclination. However, they do not specify what the inclination refers to: whether it is the inclination of the PV modules or the supports of the PV modules. The term "PV panel support" suggests that it refers to the inclination of the supports, but it should be specified what exactly is meant. Moreover, more importantly, this conclusion lacks information on whether this relationship is linear or not, and if there is a critical point. If so, do such cases occur in practical applications at all?

  2. Lines 85-87 need clarification on how the authors understand "a gradual increase in the slope angle for the single-slope PV car-shed roof" since two inclinations of 20 degrees and 30 degrees were mentioned earlier. Furthermore, similarly to the previous case, the conclusion is quite general, e.g., "With a gradual increase in the roof dip angle," which in scientific work means "a gradual increase" should specify the step of increase. And further "wind suction at the measuring point gradually increased" - gradually meaning how? With what step?

  3. Lines 98-99, the authors, by citing Naeiji's work, only state that "the most important factor influencing the peak pressure coefficients that were produced was the panel inclination angle." "that the most important factor..." suggests that other factors were also examined. From a review article, it would be worth learning what other factors were studied and to what extent they influenced the damage to PV module supports.

  4. Also, in section 2.6, the authors write who researched what, but they do not conduct a detailed analysis of the conclusions presented in already existing articles. A literature review would be valuable for analyzing "how it is in the majority" of cases.

  5. Lines 290-294, the authors write "Despite the fact that different academics have varied views on the effect of template gap on PV support wind load..." the authors believe that the gap in the template plays a significant role in determining the wind load of PV supports," which in the reviewer's opinion is not proven in this article.

  6. Lines 298-299, the authors write that Erwin and others conducted research, but they do not mention anything about the conclusions drawn from the analysis of the obtained results.

  7. Line 375 - the absorption of solar energy is not the only reason for the increase in temperature inside working photovoltaic modules. There are many publications discussing the increase in temperature inside solar cells, but there are also publications pointing out the causes of the increase in temperature inside PV panels, e.g., the publication doi.org/10.3390/en14196247.

Lines 410-411 "responses" - what kind?

There are more examples of such issues in the article submitted for review. In the reviewer's opinion, scientific work should avoid generalizations typical of populist works.

Additionally, the description contained in 468-472 does not match Figure 17.

Another issue is the nomenclature used in the article. The authors seem to interchangeably use the terms "PV module" and "PV panel." However, these words have different definitions, and it should be clarified what the authors mean.

Author Response

In view of the fact that PV support structure is vulnerable to wind load and snow load, only the effect of wind load on PV support structure is discussed in this manuscript. We quote the previous research conclusions from published papers. Under your suggestion, we have made a detailed analysis of these conclusions, and carefully studied or directly eliminated some loose and general conclusions. It is hoped that this paper can provide some useful suggestions for the wind resistance design of PV support structure, so as to improve the power generation efficiency of PV power generation system.

Comment 1. Lines 78-81, the authors cite the article by Ma et al., from which it follows that the risk of damaging PV module supports due to wind pressure increases with the angle of inclination. However, they do not specify what the inclination refers to: whether it is the inclination of the PV modules or the supports of the PV modules. The term "PV panel support" suggests that it refers to the inclination of the supports, but it should be specified what exactly is meant. Moreover, more importantly, this conclusion lacks information on whether this relationship is linear or not, and if there is a critical point. If so, do such cases occur in practical applications at all?

Response: Thanks very much for the kind suggestions. Lines 78-81 in the original text have become lines 78-84 in the revised text. "The angle of inclination" is the inclination of the angle of inclination of the PV panel; In addition, the risk of PV panel damage increases with the increase of the angle of inclination of the PV panel, this relationship is not linear, but also consider the factor of power generation efficiency, so there will be a critical point, if the panel inclination is too low, then the power generation efficiency will be affected, but the critical point value remains to be studied.

Comment 2. Lines 85-87 need clarification on how the authors understand "a gradual increase in the slope angle for the single-slope PV car-shed roof" since two inclinations of 20 degrees and 30 degrees were mentioned earlier. Furthermore, similarly to the previous case, the conclusion is quite general, e.g., "With a gradual increase in the roof dip angle," which in scientific work means "a gradual increase" should specify the step of increase. And further "wind suction at the measuring point gradually increased" - gradually meaning how? With what step?

Response: Thanks very much for the kind suggestions. Lines 85-87 in the original text have become lines 87-91 in the revised text. In the article cited by li, only several slope inclination tests of 20° and 30° were conducted, and no intermediate data tests were conducted, so the gradual increase was slightly fuzzy and it was uncertain whether the relationship was linear. Therefore, in order to be more rigorous, the conclusion is changed to that the greater the slope inclination, the greater the wind load value. Similarly, the greater the slope inclination, the greater the extreme wind suction of the measuring point.

Comment 3. Lines 98-99, the authors, by citing Naeiji's work, only state that "the most important factor influencing the peak pressure coefficients that were produced was the panel inclination angle." "that the most important factor..." suggests that other factors were also examined. From a review article, it would be worth learning what other factors were studied and to what extent they influenced the damage to PV module supports.

Response: Thanks very much for the kind suggestions. Lines 98-99 in the original text have become lines 99-103 in the revised text. Naeiji et al. studied the effects of panel inclination angle, gap height and building height on wind loads of PV roof support structures, and found that: The gap height and building height have little impact on it, the value change is less than 5%, it can be said that there is almost no impact, but the impact of the panel inclination angle is huge, the value change is 43%, so there is no mention of how much other factors affect the damage of the PV support structure, only mention the impact of the panel inclination angle on the roof PV support structure.

Comment 4. Also, in section 2.6, the authors write who researched what, but they do not conduct a detailed analysis of the conclusions presented in already existing articles. A literature review would be valuable for analyzing "how it is in the majority" of cases.

Response: Thanks very much for the kind suggestions. We have carried out a detailed analysis and systematic summary of the studies of scholars in Section 2.6, and finally concluded that the impact of template gap on PV wind load cannot be ignored. Most scholars believe that the smaller the template gap and the larger the ground clearance, the greater the wind load of PV support structure. However, the specific effect of template gap on it needs further experimental study and confirmation.

Comment 5. Lines 290-294, the authors write "Despite the fact that different academics have varied views on the effect of template gap on PV support wind load..." the authors believe that the gap in the template plays a significant role in determining the wind load of PV supports," which in the reviewer's opinion is not proven in this article.

Response: Thanks very much for the kind suggestions. Lines 290-294 in the original text have become lines 294-302 in the revised text. "Whether the template gap plays an important role in determining PV wind load" needs further experimental research, this paper directly pointed out that "the template gap plays an important role in determining PV wind load" is not rigorous, we have been corrected in the paper, but also put forward suggestions for future research.

Comment 6. Lines 298-299, the authors write that Erwin and others conducted research, but they do not mention anything about the conclusions drawn from the analysis of the obtained results.

Response: Thanks very much for the kind suggestions. Lines 298-299 in the original text have become lines 306-310 in the revised text. We have supplemented the conclusions reached by Erwin et al. and highlighted them in red.

Comment 7. Line 375 - the absorption of solar energy is not the only reason for the increase in temperature inside working photovoltaic modules. There are many publications discussing the increase in temperature inside solar cells, but there are also publications pointing out the causes of the increase in temperature inside PV panels, e.g., the publication doi.org/10.3390/en14196247.

Response: Thanks very much for the kind suggestions. The revised text has now become lines 365-367. We have fully understood the cause of the increase in temperature inside photovoltaic panels and have successfully cited your recommended article in this manuscript.

Lines 410-411 "responses" - what kind?

Response: Thanks very much for the kind suggestions. The revised text is now line 421. The responses mentioned in this manuscript are all wind-induced vibration of PV support structures. We have carried out the unification in this paper, named wind-induced vibration.

There are more examples of such issues in the article submitted for review. In the reviewer's opinion, scientific work should avoid generalizations typical of populist works.

Response: Thanks very much for the kind suggestions. In order to ensure the rigor of scientific research and avoid generalizations typical of populist works, we have carefully revised this manuscript.

Additionally, the description contained in 468-472 does not match Figure 17.

Response: Thanks very much for the kind suggestions. Lines 468-472 in the original text have become lines 476-480 in the revised text. We have re-simulated the data, and figures 16 and 17 are in complete agreement with the text description.

Another issue is the nomenclature used in the article. The authors seem to interchangeably use the terms "PV module" and "PV panel." However, these words have different definitions, and it should be clarified what the authors mean.

Response: Thanks very much for the kind suggestions. In this manuscript, "PV module" and "PV panel" both refer to "PV panel" and "PV module" has a broad meaning, and the definition in this manuscript has been unified as "PV panel".

Finally, we appreciate very much for your time in editing our manuscript and the referees for their valuable suggestions and comments. I am looking forward to hearing from your final decision when it is made.

Round 2

Reviewer 3 Report

Comments and Suggestions for Authors

Thank you to the Authors for considering the comments on the article's text. In its currently proposed form by the Authors, the article appears to be well-organized, and the issues discussed within it are comprehensively described. The literature referenced in the article is appropriate and relevant to the topics addressed.